

# D-CyPre: a machine learning-based tool for accurate prediction of human CYP450 enzyme metabolic sites

Haolan Yang[1,2], Jie Liu[2], Kui Chen[1,2], Shiyu Cong[1,2], Shengnan Cai[1,2], Yueting Li[1,2], Zhixin Jia[2], Hao Wu[1,2], Tianyu Lou[1,2], Zuying Wei[1,2], Xiaoqin Yang[1,2] and Hongbin Xiao[2]

[1] School of Chinese Materia Medica, Beijing University of Chinese Medicine, Beijing, China
[2] Beijing University of Chinese Medicine, Research Center of Chinese Medicine Analysis and Transformation, Beijing, China

## ABSTRACT

The advancement of graph neural networks (GNNs) has made it possible to accurately predict metabolic sites. Despite the combination of GNNs with XGBOOST showing impressive performance, this technology has not yet been applied in the realm of metabolic site prediction. Previous metabolic site prediction tools focused on bonds and atoms, regardless of the overall molecular skeleton. This study introduces a novel tool, named D-CyPre, that amalgamates atom, bond, and molecular skeleton information *via* two directed message-passing neural networks (D-MPNN) to predict the metabolic sites of the nine cytochrome P450 enzymes using XGBOOST. In D-CyPre Precision Mode, the model produces fewer, but more accurate results (Jaccard score: 0.497, F1: 0.660, and precision: 0.737 in the test set). In D-CyPre Recall Mode, the model produces less accurate, but more comprehensive results (Jaccard score: 0.506, F1: 0.669, and recall: 0.720 in the test set). In the test set of 68 reactants, D-CyPre outperformed BioTransformer on all isoenzymes and CyProduct on most isoenzymes (5/9). For the subtypes where D-CyPre outperformed CyProducts, the Jaccard score and F1 scores increased by 24% and 16% in Precision Mode (4/9) and 19% and 12% in Recall Mode (5/9), respectively, relative to the second-best CyProduct. Overall, D-CyPre provides more accurate prediction results for human CYP450 enzyme metabolic sites.

## INTRODUCTION

Drug metabolism is closely linked to the bioavailability, bioactivity, and toxicology of those drugs. Cytochrome P450 (CYP450) enzymes are responsible for the metabolism of approximately 90% of FDA-approved medicines and play a vital role in the Phase I metabolism of drugs (*Nebert & Russell, 2002*). Humans have more than 50 CYP450 isozymes, of which CYP1A2, CYP2A6, CYP2B6, CYP2C8, CYP2C9, 2C19, CYP2D6, CYP2E1, and CYP3A4 are responsible for the Phase I metabolism of most known drugs (*Nebert & Russell, 2002*; *Furge & Guengerich, 2006*), so predicting drug metabolism by CYP450 isoforms is crucial for drug design and discovery (*Jianing et al., 2011*).

Corresponding author
Hongbin Xiao, hbxiao69@163.com

Several *in silico* metabolism prediction tools have been developed, such as CyProduct (*Tian et al., 2021*), CypReact (*Tian et al., 2018*), FAME2 (*Šícho et al., 2017*), FAME3 (*Šícho et al., 2019*), and BioTransformer (*Djoumbou-Feunang et al., 2019*). However, all these models rely on fixed rules to generate the features of the site of metabolism (SOM) or bond of metabolism (BOM). Although graph neural networks (GNNs) are less prevalent *in silico* metabolism prediction tasks, they have already demonstrated their efficacy in replacing conventionally handcrafted molecular features generated by fixed rules in other molecular-related research domains. Recently, GNNs have shown promise in molecular property prediction (*Gilmer et al., 2017*; *Yang et al., 2019*) and drug discovery (*Stokes et al., 2020*; *Jin et al., 2021*). The GNNs used in these studies were message-passing neural networks (MPNN; *Gilmer et al., 2017*; *Jo et al., 2020*) and directed MPNN (D-MPNN; *Yang et al., 2019*; *Stokes et al., 2020*; *Jin et al., 2021*; *Han et al., 2022*). Both of these networks use message-passing to aggregate the chemical information from the entire molecule and learn how to generate better features. The difference between these two networks lies in the types of messages they aggregate: MPNN aggregates information from related vertices (atoms), while D-MPNN aggregates information from directed edges (bonds). Compared to the MPNN, the D-MPNN can avoid loops in message-passing (*Yang et al., 2019*).

In many studies predicting SOMs or BOMs, the models often include information about neighboring atoms or bonds when creating features for atoms or bonds (*He et al., 2016*; *Šícho et al., 2017*, *2019*; *de Bruyn Kops et al., 2019*, *2021*; *Tian et al., 2021*). However, this step is very subjective, and it is difficult to determine which features of adjacent structures are required by the model, so there is room for improvement in models that rely on these features. In contrast, the D-MPNN requires only the features of the target atom or bond, and the neural network determines which features of neighboring structures are important. Also, the neural network does not just screen the features, but transforms the features, which may generate some new features that are more effective for determining SOMs. The D-MPNN has shown excellent results in other fields and is able to objectively and powerfully generate features. Because of this, D-MPNN may achieve better results than existing *in silico* metabolism prediction models.

A previous study thoroughly investigated the effectiveness of GNNs in predicting metabolic sites (*Porokhin, Liu & Hassoun, 2023*), but the GNN that was scrutinized does not incorporate the novel D-MPNN and has not evolved into a user-friendly tool for scientific researchers. Training stable models for molecular property prediction using a multi-layer perceptron may prove to be challenging. One study indicates that employing a GNN in conjunction with XGBOOST for training yields superior predictive performance (*Deng et al., 2021*). Furthermore, the overall structure of the molecule is a crucial factor. This study also examines the impact of fusing traditional molecular features, or features generated based on D-MPNN, with those generated from the bonds and atoms within the molecule using D-MPNN. This study introduces a novel metabolic site prediction model with better performance that could aid non-computational personnel in their research within the field of metabolism.

This study introduces D-CyPre, an *in silico* metabolism predictor capable of predicting any of the nine most significant human CYP450 enzymes (Phase I metabolism; *Zanger &*

*Schwab, 2013*). As shown in Fig. 1, D-CyPre can be divided into two parts. The first part generates the features using D-MPNN, and the second part predicts metabolic sites using these features. Finally, D-CyPre visually displays the predicted results (Fig. 1). The darker the red in the figure, the higher the probability of metabolism at this site. Additionally, the probability value is written on the target atom or bonding atom of the target bond. D-CyPre only displays sites that have a probability of metabolism greater than 50%.

# MATERIALS AND METHODS

## Data sets

The study used the EBoMD data set from CyProduct for the training set (*Tian et al., 2021*). The EBoMD data set is a public data set that includes BOMs of 679 substrates on nine of the most important human CYP450 isoforms (CYP1A2, CYP2A6, CYP2B6, CYP2C8, CYP2C9, CYP2C19, CYP2D6, CYP2E1, CYP3A4) created from the Zaretzki data set (*Zanger & Schwab, 2013*; *Zaretzki, Matlock & Swamidass, 2013*). The Zaretzki data set has been used in several related studies of *in silico* metabolism predictors (*Tian et al., 2018*, *2021*; *Šícho et al., 2019*; *Dang et al., 2020*). *Tian et al. (2021)* converted SOMs in the Zaretzki data set into BOMs during the creation of the EBoMD, while correcting some errors. The EBoMD mainly consists of the following nine Phase I reactions: Oxidation, Cleavage, EpOxidation, Reduction, Hydroxylation, S(sulfur)-Oxidation, N(nitrogen)-Oxidation, P(phosphorus)-Oxidation, and Cyclization (*Tian et al., 2021*).

EBoMD2 was used as a test data set to evaluate D-CyPre's performance and compare it with CyProduct's performance. EBoMD2 comes from CyProduct and contains 68 extracted reactants and 30 known non-CYP450 reactants (*Tian et al., 2021*). D-CyPre's accuracy in metabolite prediction was then compared to BioTransformer and CyProduct (*Djoumbou-Feunang et al., 2019*; *Tian et al., 2021*), which are predictive models for metabolites.

## Atoms and bonds of metabolism

The BOM used in this study was determined by CyProduct. *Tian et al. (2021)* argue that BOM is more clearly defined and classified more systematically than SOM. Because D-CyPre is structured differently than other models, a new definition of SOM was created based on the BOM. In D-Cypre, the features of atoms and bonds both descend to the same dimensions by neural networks, so common features of atoms and bonds can be identified. This study still refers to these defined atoms and bonds as SOMs. The specific rules of SOMs that D-CyPre should recognize are described, as follows:

(1) i-j: i and j represent any two non-H atoms currently connected by an existing chemical bond. The bond formed by these two atoms is a SOM that D-CyPre should recognize.

(2) i-H: i represents any non-H atom, and hydrogen atoms on i are replaced with heteroatoms. Atom i and the bond formed between i and H are both SOMs because this reaction involves both i and its bonds with H.

(3) SPN: When new bonds are generated on S, P, or N by sharing their lone pair of electrons, these atoms are defined as SOMs because this reaction only involves atoms (S, P, or N).

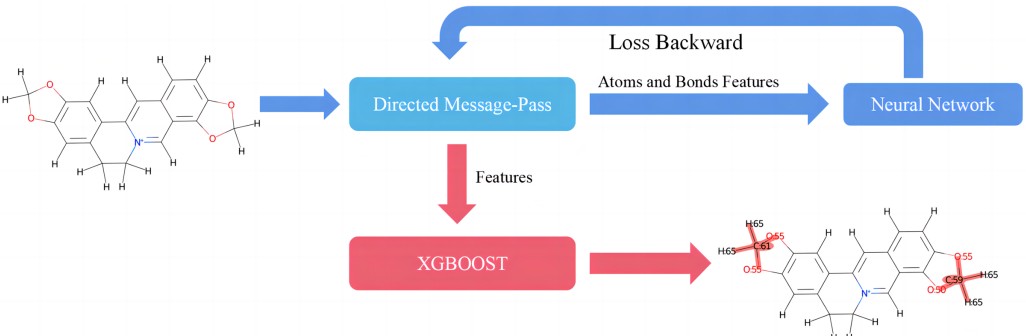

**Figure 1 Overview of D-CyPre metabolism prediction suite (shown for a specific instance of CYP2A6).**

Instead of creating a model for each type of bond, as CyProduct does (*Tian et al., 2021*), only one model was used to identify all the types of SOMs of one CYP450 isoform. D-CyPre does not treat atoms and bonds separately, but uses the same discriminator for both to determine whether they are SOMs. Atoms and bonds are determined using the same model because their associated information is combined during the message-passing process of D-MPNN, and the model is likely to learn more positive information without distinguishing between them. The distribution of SOMs for nine CYP450 isoforms is shown in Table 1 and Fig. 2. In both the EBoMD and the EBoMD2, 3A4 had the largest number of SOMs and non-SOMs, and there were large differences in the number of SOMs and non-SOMs for each isozyme.

## Generation of metabolites based on SOMs

Metabolites were generated based on whether the SOM was an atom or a bond. The detailed SMIRKS Reaction can be found in the Reaction rules of the CyProduct project (https://bitbucket.org/wishartlab/cyproduct/src/master/; *Tian et al., 2021*). The generation rules are as follows:

(1) Atom: When the SOM is an atom, oxidation occurs if the atom is S, N, or P. When the atom is C in the benzene ring and is connected with H, hydroxylation should occur. If the C is not on the benzene ring and there are N or O atoms, all the bonds connected to the N and O atoms are broken. If the molecule is broken into two products, the C of the product will further oxidize. If the molecule is still one product, the metabolism is not considered.

(2) Bond: If the bond is C-H and C is connected to O or N, the metabolism is based on the C atom as the SOM. Otherwise, the C-H is hydroxylated. If the atoms at both ends of the bond are C and the bond is a double bond, epoxidation occurs. If it is C-O, oxidation occurs. If it is i-O, where i is S, N, or P, the i-O is broken. If there are multiple O and i is S or P, one of the i-O is broken, and if there are multiple O and i is N, i-O is reduced to a primary amine group. If all the bonds in a ring are SOMs, then the aromatic and non-aromatic ring interconversion occurs.

**Table 1 Distribution of SOMs for nine CYP450 isoforms in data sets.**

| Data set | type | 1A2 | 2A6 | 2B6 | 2C8 | 2C9 | 2C19 | 2D6 | 2E1 | 3A4 |
|---|---|---|---|---|---|---|---|---|---|---|
| EBoMD | Reactants | 279 | 109 | 149 | 147 | 237 | 221 | 282 | 144 | 474 |
| | SOMs | 1,847 | 615 | 830 | 906 | 1,372 | 1,368 | 1,685 | 863 | 3,139 |
| | Non-SOMs | 18,760 | 5,951 | 9,914 | 11,322 | 17,481 | 16,387 | 21,596 | 7,458 | 43,597 |
| EBoMD2 | Reactants | 16 | 10 | 11 | 9 | 13 | 13 | 24 | 10 | 41 |
| | SOMs | 64 | 49 | 31 | 49 | 64 | 51 | 158 | 48 | 236 |
| | Non-SOMs | 1,182 | 631 | 596 | 946 | 1,134 | 1,180 | 2,581 | 258 | 3,788 |

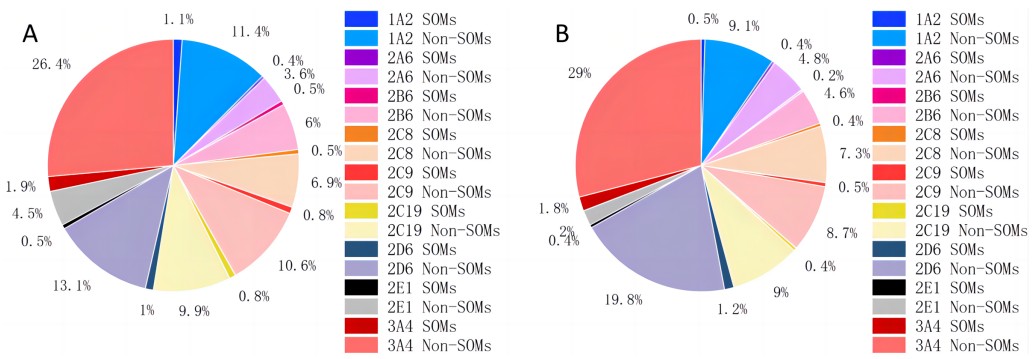

**Figure 2 Distribution of SOMs for nine CYP450 isoforms in EBoMD (A) and EBoMD2 (B).**

## Feature generation

D-CyPre includes nine atom descriptors and four bond descriptors (Table 2). Details of these descriptors are available in Table S1. The data used for training and testing D-CyPre included C, H, O, N, S, and P. To avoid producing a large number of dimensions that could be learned, the same value was assigned to all other types of atoms when calculating the atomic number.

## D-CyPre

D-CyPre consists of D-MPNN and XGBOOST, where D-MPNN outputs the features of atoms and bonds, and XGBOOST identifies SOMs based on these features. The details of these structures are as follows:

### D-MPNN

D-MPNN was built based on ComboNet's MPN, which originally came from the open-source Chemprop Software (*Yang et al., 2019*; *Jin et al., 2021*) available at https://github.com/chemprop/chemprop. First, the information about the directed bonds and their starting atoms are fused:

$$\tau(x) = \max(0, x) \tag{1}$$

$$h_{vw}^0 = \tau(W_a[x_v, e_{vw}]) \tag{2}$$

**Table 2 Descriptors of atoms and bonds.**

| Atom descriptors | Bond descriptors |
| --- | --- |
| Atomic number | Bond type (Single/Double/Triple/Aromatic) |
| Degree | Conjugation |
| Formal charge | Ring membership |
| Chirality | Stereochemistry |
| Number of bonded hydrogens | (–) |
| Hybridization | (–) |
| Aromaticity | (–) |
| Ring membership | (–) |
| Atomic mass | (–) |

where $W_a \in \mathbb{R}^{h \times h_a}$ is a learned matrix, $[x_v, e_{vw}] \in \mathbb{R}^{h_a}$ splice together $e_{vw}$, the feature of a directed bond, and $x_v$, the feature of the initial atom of the bond. To avoid a message loop through a directed bond, the features of directed bonds vw starting from atom v are characterized by concatenating the features of atom v and undirected bonds vw, and then updated through $W_a$ Eq. (2). $\tau$ is the LeakyReLU activation function (*Xu et al., 2015*). Then, the message-passing begins:

$$m_{vw}^{t+1} = \sum_{\{k|k \in N(v), k \neq w\}} h_{kv}^t \tag{3}$$

$$h_{vw}^{t+1} = drop\left(\tau\left(h_{vw}^0 \oplus W_m m_{vw}^{t+1}\right)\right) \tag{4}$$

where $W_m \in \mathbb{R}^{h \times h}$ is a learned matrix, *drop* is the Dropout layer (*Srivastava et al.*), and $N(v)$ represents all the atoms connected to v. The features surrounding the directed bond kv are first aggregated through Eq. (3). The features aggregated in Eq. (3) are then updated through $W_m$ in Eq. (4) and are combined with the original features of the directed bond kv. This is because the initial features are the most important information directly related to the directed bond kv. Finally, the features of the directed bond kv are obtained after one message-passing action. The message-passing action is repeated *n* times, which represents the depth of the message-pass; the greater the *n*, the farther the message will pass. Then, the features of the bonds and atoms are calculated from the message:

$$F_{vw} = \mathcal{B}\left(\frac{h_{vw}^n \oplus h_{wv}^n}{2}\right) \tag{5}$$

$$F_v = \mathcal{B}\left(drop\left(\tau\left(W_o\left[x_v, \sum_{W \in N(v)} h_{vw}^n\right]\right)\right)\right) \tag{6}$$

where $\mathcal{B}$ is the Batch Normalization (*Ioffe & Szegedy, 2015*), $W_o \in \mathbb{R}^{h \times h_b}$ is a learned matrix. Note that the same Batch Normalization layer is used for both atoms and bonds. When calculating the features of the bond formed by atoms v and w, the mean of the bond features in both directions, after multiple message-passing actions, is calculated and normalized by Batch Normalization Eq. (5). To calculate the features of atom v, all the

features of the directed bond starting from the atom v are gathered and spliced with the initial features of the atom. This is because the initial features of the atom are the closest features to the atom's own state. The features are then updated based on $W_o$, and the final features are obtained through activation functions Dropout and Batch Normalization Eq. (6). Then, $F_v$ and $F_{vw}$ are fed into a single-layer neural network, and two values are obtained for each bond and atom: the positive probability and the negative probability. The cross-entropy is then used to calculate the loss of the model:

$$loss = a \times loss_p + b \times loss_n \tag{7}$$

where $loss_p$ and $loss_n$ are the loss of atoms and bonds that are truly labeled positive and negative, respectively; and $a$ and $b$ are two self-defined parameters, which respectively represent the importance attached to $loss_p$ and $loss_n$. These two parameters are adjusted when training models of the different CYP450 isoforms. This approach is adopted because the data set is unbalanced, with a large number of negative data and only a small number of positive data (Fig. 2). Because of this imbalance, the model may classify all samples as negative resulting in an extremely low loss. Therefore, by setting the loss value in this manner, the weight of positive and negative data can be flexibly adjusted while treating them equally. This ensures that the recall rate of the model is not too low.

### XGBOOST

XGBOOST was introduced by *Chen & Guestrin (2016)* and has demonstrated excellent results in several studies (*Yu et al., 2019*; *Chen et al., 2021*; *Zhang, Hu & Yang, 2022*). *Deng et al. (2021)* showed that the D-MPNN + XGBOOST model can effectively improve the prediction of various molecular properties compared to D-MPNN alone. Therefore, this study used D-MPNN + XGBOOST training models. In general, an XGBOOST model was trained based on $F_{vw}$ and $F_v$ and output Jaccard score (TP/(TP + FP + FN)), precision (TP/(TP + FP)), recall (TP/(TP + FN)), and F1 ($2 \times$ precision $\times$ recall/(precision + recall)) in each epoch of D-MPNN. D-MPNN updates the feature generator and the single-layer neural network based on the loss value. The updated feature generator generates features that are passed to XGBOOST to train the real discriminator. The objective and Feval for XGBOOST are set to "binary: logistic" and Jaccard score, respectively. Other parameters for XGBOOST such as "n estimators", "reg lambda", "max depth" and "colsample bytree" are tuned for different isoforms.

### Molecular features

Molecular features play a crucial role in identifying SOMs. For instance, two atoms or bonds in similar conditions may react differently with CYP450 due to their molecular structure: one may react, while the other may not. Such bonds or atoms are hard to identify without including molecular features. This study considered two types of molecular features: the first type was generated by a new D-MPNN (*Yang et al., 2019*), and the second type was directly calculated according to specific rules (MolWt; NumHAcceptors; NumHDonors; MolLogP; TPSA; LabuteASA). Details of these descriptors can be found in Table S1.

The molecular features in this study were directly concatenated with the features of the atoms and bonds contained in that molecule. Although molecular features are important, their introduction did not necessarily improve the Jaccard Score of all models in this study. There are two main reasons for this. First, the model used in this study is already complex, so introducing molecular features might not have been able to further improve it and could have even caused more severe overfitting. Second, because the data set was not large enough, the model could only learn a small amount of molecular information, which may have been a disturbance for some isoforms of CYP450. Figure 3 illustrates the structure of D-CyPre that incorporates molecular features.

### Precision mode and recall mode

D-CyPre has two modes: high precision and high recall. The difference between the two is that in Precision Mode, XGBOOST's "scale pos weight" is set to the default, while in Recall Mode, this parameter is set to (c × Positive/Negative), where c is a parameter that can be adjusted.

### Training model

For any CYP450 isoform, the EBoMD was divided into a training set and a validation set, in a ratio of 8:2 (since the features of SOMs are affected by the entire molecular structure, molecules were used, rather than SOMs, as the minimum unit when dividing the data set). Based on these data, the model parameters were adjusted to obtain those with a high Jaccard score in both the training and validation sets.

During this process, D-MPNN was trained using data from all isoforms and then XGBOOST was trained using data from only the target isoform (Fig. 4). This improved both the Jaccard score and the generalization ability of the model because the metabolism information of nine isoforms had some overlap. Although learning more knowledge from other isoforms may introduce some noise into the model, this knowledge and moderate noise enhance the model's generalization ability (File S1). The same method, based on parameters with a high Jaccard score in both training and validation sets, was used to train the final D-CyPre and test it with the test set.

## EXPERIMENTALS AND RESULTS

### The results of training based on SOMs

The result of CypBOM is a cross-validation result of CyProduct in predicting metabolic sites, which is only used as a simple reference because of slight differences in the definition of metabolic sites. For metabolite prediction, after optimizing the SOMs prediction model, it is more valuable to convert the SOMs predicted by the model into metabolites and compare them with CyProduct and BioTransformer (the results of testing based on metabolites).

### Training results of precision mode

Training results are shown in Table S2. The Jaccard score of D-CyPre-val for nine CYP450 enzymes was higher than the CypBoM Jaccard Score for these enzymes. Similarly, D-CyPre-val showed higher precision and F1 values for all CYP450 enzymes except 2C8.
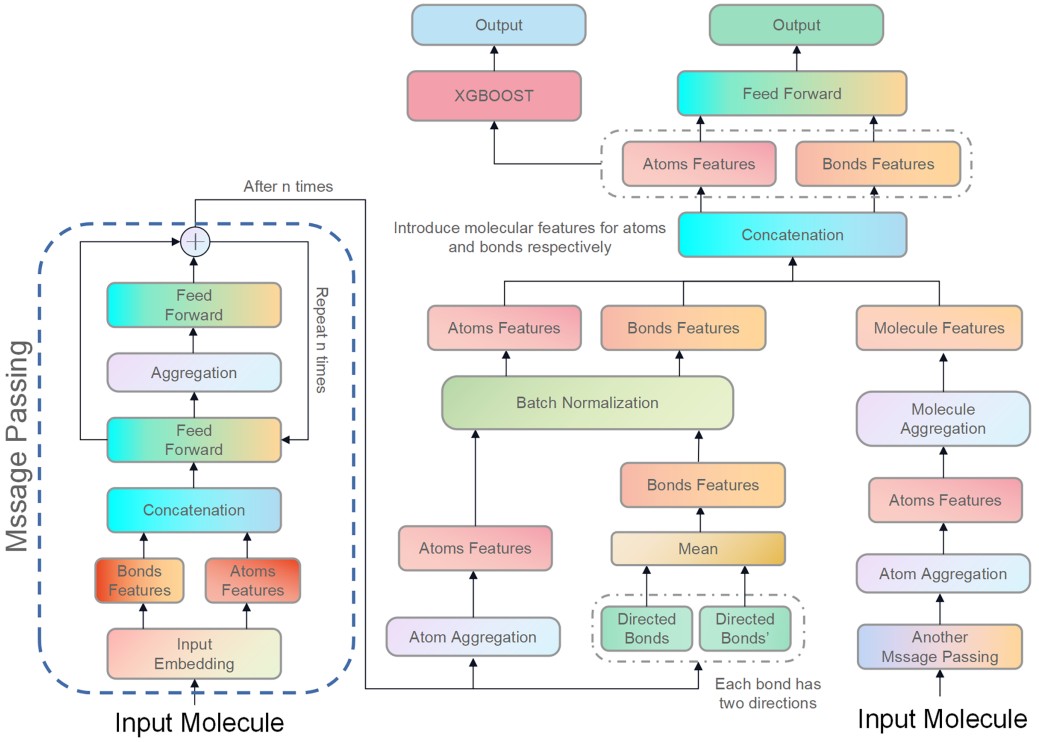

**Figure 3 Illustration of the proposed model, D-CyPre.** D-CyPre employs two independent message-passing processes to capture features of two kinds of directed bonds from a molecule. It then fuses the features of the two kinds of directed bonds to derive features of atoms, chemical bonds, and molecules. The features of atoms and bonds are separately combined with those of the molecule and input into a feed forward layer to generate prediction probabilities, which then update the network. The concatenated features of atoms and bonds are then fed into the XGBOOST model to obtain the actual prediction probabilities.

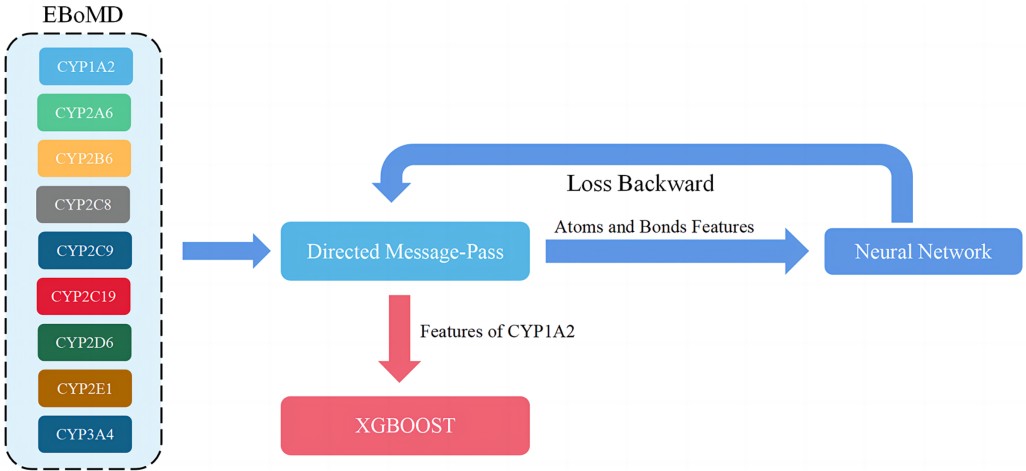

**Figure 4 Overview of the training model (shown for a specific instance of CYP1A2).** When adjusting parameters, only the training set (80% of EBoMD) of 1A2 and all data sets (100% of EboMD) of the other isoforms were used to train the model. All training sets (100% of EBoMD) of 1A2 and the other isoforms (100% of EBoMD) were used when training the final model.

However, because D-CyPre-Val and CypBoM use different validation sets and methods, this result does not prove that D-CyPre has a better predictive power than CypBoM.

### Training results of recall mode

According to results shown in Table S3, D-CyPre-val produced a higher Jaccard score, recall, and F1 for nine CYP450 enzymes. However, D-CyPre-val failed to achieve better precision in the data of 1A2, 2B6, and 2C8 under Recall Mode. The weighted average of all metrics of D-CyPre-val was higher than that of CypBoM. This indicates that D-CyPre has good predictive power. In Recall Mode, D-CyPre produces comprehensive results while retaining high precision.

## The results of testing based on SOMs

### Testing results of precision mode

Using Precision Mode, D-CyPre enhanced precision (WAvg) by 1,136% on the test set compared to Random Predictor (Table S4). Compared to CyProduct, D-CyPre also produced higher Jaccard score (WAvg), recall (WAvg), and F1 (WAvg) values, with increases of 800%, 21%, and 535%, respectively. The results from the training set are displayed in Table S2, with D-CyPre exhibiting exceptionally high precision values for several of the nine CYP450 enzymes in both the training set and test set. For example, the precision values for 2A6 and 2E1 in the training and test sets surpassed 0.8 and 0.9, respectively.

Unfortunately, D-CyPre in Precision Mode did not exhibit strong performance across all enzymes. Despite the fact that D-CyPre performed well for 2B6 and 2C8 in the validation set (Table S2), their results in the test set indicated severe overfitting (Table S4). CyProduct also encountered this issue (*Tian et al., 2021*), with the models for 2B6 and 2C8 performing well in the validation set but performing poorly in the test set. To further investigate the underlying causes of this problem, t-SNE was used to visualize the SOMs based on features generated by D-MPNN (*van der Maaten & Hinton, 2008*). The SOMs in the training set, validation set, and test set of these models were all visualized. The green box in Fig. 5A represents potential false negatives in the test set that reduced Recall for the 2B6 model. Similarly, Figs. 5C and 5D illustrate possible sources of error for 2C8. From these results, it can be inferred that there may be two reasons for the poor generalization ability of these models on the test set. First, an insufficiently sized training set might have led to some bonds or atoms in the test set that are similar in structure to SOMs being incorrectly classified as positive (Figs. 5A and 5C), as shown in the part of the graph where blue + coincides with red +. Some unfamiliar actual SOMs are incorrectly classified as negative (Figs. 5B and 5D), as shown in the part of the graph where red + coincides with blue +. The second possible source of error is that 2B6 and 2C8 had almost the largest ratios of Non-SOMs/SOMs (Table S5) in their respective test sets, which may have made Precision more sensitive to errors. On larger test sets, the test results of the model might perform more closely to the validation set. Furthermore, neither the test nor validation sets were distributed within regions lacking training data, indicating that there were no atoms or bonds present in either set that had not been previously encountered by the models.

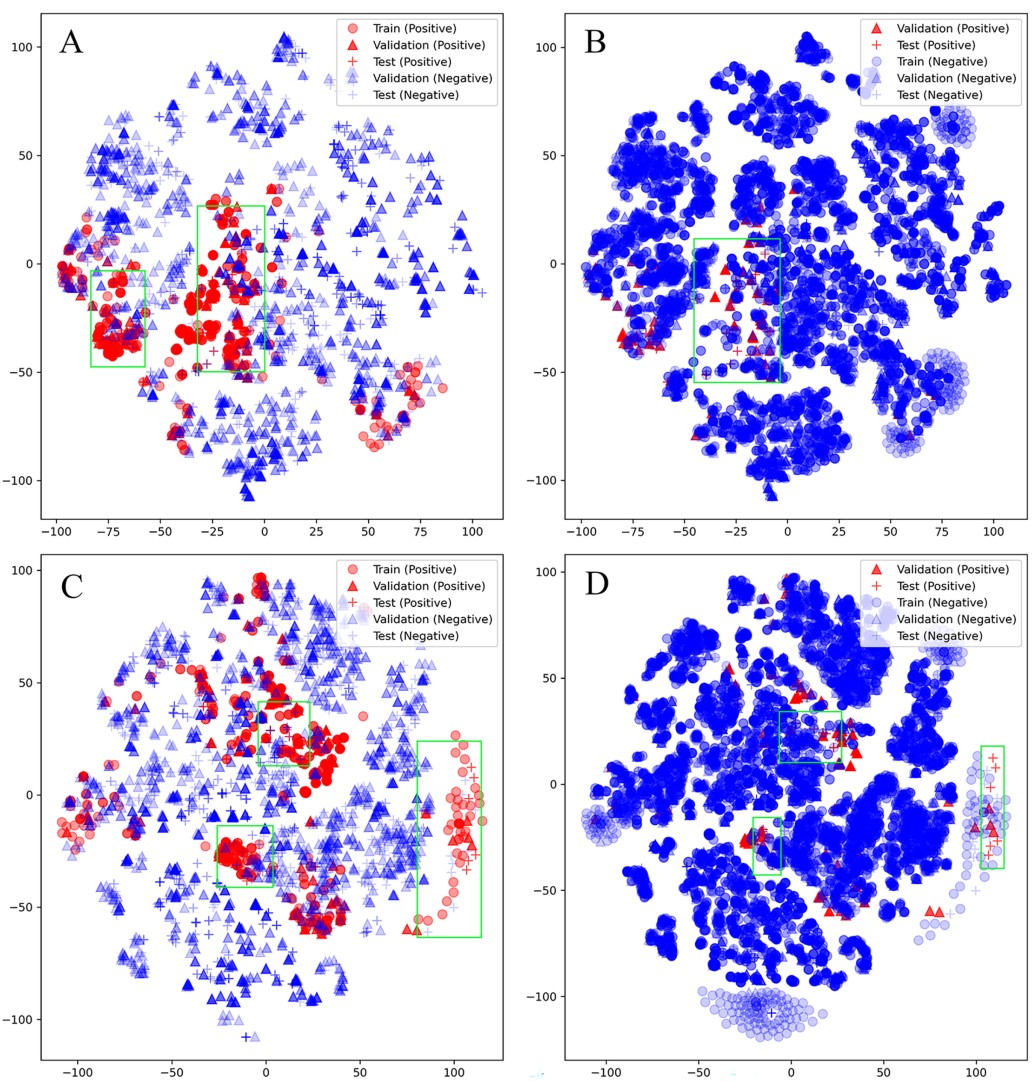

**Figure 5 Visualization (by t-SNE) of the SOMs of 2B6 and 2C8.** Visualization (by t-SNE) of the SOMs of 2B6 (ignore training set; negative) (A), 2B6 (ignore training set; positive) (B), 2C8 (ignore training set; negative) (C) and 2C8 (ignore training set; positive) (D). The green box part is some data that the model may misjudge.

This indicates that D-CyPre's chemical space based on its training data was sufficiently large.

### Testing results of recall mode

As shown in Table S6, compared to Random Predictor, D-CyPre exhibited an 813% increase in Jaccard score (WAvg), a 981% increase in Precision (WAvg), a 44% increase in Recall (WAvg), and a 545% increase in F1 (WAvg). Additionally, the Jaccard scores for 2B6 and 2C8 also improved under Recall Mode. Overall, D-CyPre successfully maintained high Jaccard scores while achieving high recall.

**Table 3 Training results (Precision Mode) for nine CYP450 enzymes in EBoMD and EBoMD2.**

| | 1A2 | 2A6 | 2B6 | 2C8 | 2C9 | 2C19 | 2D6 | 2E1 | 3A4 | WAvg[e] |
|---|---|---|---|---|---|---|---|---|---|---|
| | Jaccard score TP/(TP + FP + FN) | | | | | | | | | |
| D-CyPre[a] | 0.845 | 0.919 | 0.625 | 0.832 | 0.680 | 0.650 | 0.545 | 0.728 | 0.791 | 0.733 |
| D-CyPre-val[b] | 0.475 | 0.695 | 0.489 | 0.500 | 0.512 | 0.573 | 0.550 | 0.703 | 0.469 | 0.527 |
| D-CyPre-all[c] | 0.826 | 0.880 | 0.644 | 0.760 | 0.685 | 0.660 | 0.560 | 0.744 | 0.765 | 0.722 |
| D-CyPre-test[d] | 0.593 | 0.549 | 0.333 | 0.281 | 0.639 | 0.492 | 0.548 | 0.469 | 0.462 | 0.497 |
| | Precision TP/(TP + FP) | | | | | | | | | |
| D-CyPre | 0.989 | 0.994 | 0.896 | 0.990 | 0.831 | 0.721 | 0.747 | 0.860 | 0.968 | 0.891 |
| D-CyPre-val | 0.832 | 0.953 | 0.830 | 0.729 | 0.758 | 0.662 | 0.843 | 0.867 | 0.769 | 0.792 |
| D-CyPre-all | 0.978 | 0.998 | 0.702 | 0.983 | 0.745 | 0.751 | 0.759 | 0.876 | 0.962 | 0.871 |
| D-CyPre-test | 0.699 | 0.933 | 0.500 | 0.667 | 0.852 | 0.750 | 0.735 | 0.958 | 0.676 | 0.737 |
| | Recall TP/(TP + FN) | | | | | | | | | |
| D-CyPre | 0.854 | 0.924 | 0.674 | 0.839 | 0.790 | 0.869 | 0.668 | 0.825 | 0.812 | 0.802 |
| D-CyPre-val | 0.526 | 0.719 | 0.543 | 0.614 | 0.613 | 0.811 | 0.613 | 0.787 | 0.546 | 0.618 |
| D-CyPre-all | 0.841 | 0.881 | 0.887 | 0.770 | 0.896 | 0.844 | 0.682 | 0.832 | 0.788 | 0.812 |
| D-CyPre-test | 0.797 | 0.571 | 0.500 | 0.327 | 0.719 | 0.588 | 0.684 | 0.479 | 0.593 | 0.610 |
| | F1 2 × Precision × Recall/(Precision + Recall) | | | | | | | | | |
| D-CyPre | 0.917 | 0.958 | 0.769 | 0.908 | 0.810 | 0.788 | 0.705 | 0.842 | 0.883 | 0.841 |
| D-CyPre-val | 0.645 | 0.820 | 0.657 | 0.667 | 0.678 | 0.729 | 0.710 | 0.825 | 0.639 | 0.688 |
| D-CyPre-all | 0.904 | 0.936 | 0.784 | 0.864 | 0.814 | 0.795 | 0.718 | 0.853 | 0.866 | 0.835 |
| D-CyPre-test | 0.745 | 0.708 | 0.500 | 0.439 | 0.780 | 0.659 | 0.709 | 0.639 | 0.632 | 0.660 |

**Notes:**
[a] The results of D-CyPre on the training set.
[b] The results of D-CyPre on the validation set.
[c] The results of D-CyPre on EBoMD.
[d] The results of D-CyPre on EBoMD2.
[e] The microaverage (weighted average, weighted by the number of SOMs) over the nine CYP450 enzymes.

## The results of training D-CyPre with the molecular features on SOMs

The effects of two molecular features on 1A2 and 2B6 (Table S7) were compared and the results showed that the molecular features calculated by D-MPNN exhibited some advantages over those calculated using fixed rules. As a result, the same methodology was applied to construct a version of D-CyPre that incorporated molecular features calculated by D-MPNN. However, this version of D-CyPre did not exhibit better performance across all isoforms when compared with the original version of D-CyPre (Tables S8 and S9). Subsequently, optimal models from both versions of D-CyPre (with or without molecular features) were synthesized to obtain a new Precision Mode (Table 3) and Recall Mode (Table 4); 1A2, 2A6, 2B6, 2C8, 2C9, and 2C19 enzymes were all adopted by models incorporating molecular features under both modes, which suggests that molecular structure may be an important factor affecting metabolic reactions for these enzymes. The parameters for loss function and XGBOOST for all models can be found in Table S10.

### Training results of precision mode

In Precision Mode (Table S8), the results of the training set showed that the D-CyPre-val of 1A2, 2A6, 2B6, 2C8, 2C9, 2C19, 2E1, and 3A4 enzymes all improved after the

**Table 4 Training results (Recall Mode) for nine CYP450 enzymes in EBoMD and EBoMD2.**

| | 1A2 | 2A6 | 2B6 | 2C8 | 2C9 | 2C19 | 2D6 | 2E1 | 3A4 | WAvg[e] |
|---|---|---|---|---|---|---|---|---|---|---|
| | Jaccard score TP/(TP + FP + FN) | | | | | | | | | |
| D-CyPre[a] | 0.872 | 0.915 | 0.644 | 0.788 | 0.688 | 0.835 | 0.575 | 0.729 | 0.646 | 0.723 |
| D-CyPre-val[b] | 0.501 | 0.708 | 0.577 | 0.504 | 0.554 | 0.619 | 0.561 | 0.709 | 0.517 | 0.561 |
| D-CyPre-all[c] | 0.842 | 0.907 | 0.644 | 0.774 | 0.685 | 0.829 | 0.588 | 0.742 | 0.656 | 0.722 |
| D-CyPre-test[d] | 0.571 | 0.636 | 0.358 | 0.365 | 0.580 | 0.463 | 0.554 | 0.500 | 0.468 | 0.506 |
| | Precision TP/(TP + FP) | | | | | | | | | |
| D-CyPre | 0.970 | 0.965 | 0.689 | 0.848 | 0.752 | 0.923 | 0.636 | 0.820 | 0.728 | 0.799 |
| D-CyPre-val | 0.798 | 0.934 | 0.652 | 0.702 | 0.678 | 0.748 | 0.664 | 0.830 | 0.657 | 0.719 |
| D-CyPre-all | 0.957 | 0.956 | 0.702 | 0.840 | 0.745 | 0.920 | 0.643 | 0.837 | 0.751 | 0.803 |
| D-CyPre-test | 0.658 | 0.854 | 0.463 | 0.519 | 0.734 | 0.660 | 0.615 | 1.000 | 0.570 | 0.645 |
| | Recall TP/(TP + FN) | | | | | | | | | |
| D-CyPre | 0.897 | 0.946 | 0.907 | 0.918 | 0.890 | 0.897 | 0.857 | 0.868 | 0.852 | 0.882 |
| D-CypPe-val | 0.574 | 0.746 | 0.833 | 0.641 | 0.752 | 0.781 | 0.783 | 0.830 | 0.708 | 0.725 |
| D-CyPre-all | 0.875 | 0.946 | 0.887 | 0.907 | 0.896 | 0.894 | 0.872 | 0.868 | 0.838 | 0.876 |
| D-CyPre-test | 0.813 | 0.714 | 0.613 | 0.551 | 0.734 | 0.608 | 0.848 | 0.500 | 0.725 | 0.720 |
| | F1 2 × Precision × Recall/(Precision + Recall) | | | | | | | | | |
| D-CyPre | 0.932 | 0.955 | 0.783 | 0.882 | 0.815 | 0.910 | 0.730 | 0.843 | 0.785 | 0.835 |
| D-CyPre-val | 0.668 | 0.829 | 0.731 | 0.670 | 0.713 | 0.764 | 0.719 | 0.830 | 0.682 | 0.717 |
| D-CyPre-all | 0.914 | 0.951 | 0.784 | 0.872 | 0.814 | 0.907 | 0.740 | 0.852 | 0.792 | 0.835 |
| D-CyPre-test | 0.727 | 0.778 | 0.528 | 0.535 | 0.734 | 0.633 | 0.713 | 0.667 | 0.638 | 0.669 |

**Notes:**
[a] The results of D-CyPre on train set.
[b] The results of D-CyPre on validation set.
[c] The results of D-CyPre on EBoMD.
[d] The results of D-CyPre on EBoMD2.
[e] The microaverage (weighted average, weighted by the number of SOMs) over the nine CYP450 enzymes.

introduction of molecular features. The Jaccard score of subtype 3A4 improved the most, increasing by 5.4% after the introduction of molecular features. The Jaccard score of seven subtypes increased by an average of 3.1% after the introduction of molecular features.

### Training results of recall mode

In Recall Mode (Table S9), the D-CyPre-val of 1A2, 2A6, 2B6, 2C8, 2C9, and 2C19 enzymes improved after the introduction of molecular features. The Jaccard Score of subtype 1A2 exhibited the greatest improvement, with a notable increase of 9.2% upon the introduction of molecular features. On average, the Jaccard score of the four subtypes showed an increase of 6.5% following the incorporation of molecular features.

## Testing results of D-CyPre with the molecular features on SOMs
### Testing results of precision mode

Compared with the Random Predictor, the D-CyPre-test increased Jaccard score (WAvg) by 820%, precision (WAvg) by 1,171%, recall (WAvg) by 22%, and F1 (WAvg) by 606%. Compared to the D-CyPre-test without molecular features, the D-CyPre-test with

molecular features saw a 2% increase in Jaccard score (WAvg), a 3% increase in precision (WAvg), a 1% increase in recall (WAvg), and a 2% increase in F1 (WAvg).

### Testing results of recall mode

Compared with the Random Predictor, D-CyPre-test increased Jaccard score (WAvg) by 837%, precision (WAvg) by 1,012%, recall (WAvg) by 44%, and F1 (WAvg) by 556%. Compared to the D-CyPre-test without molecular features, the D-CyPre-test with molecular features increased Jaccard score (WAvg) by 3%, precision (WAvg) by 3%, and F1 (WAvg) by 2%. However, there was no significant difference in recall (WAvg) compared to before the introduction of molecular features.

## The results of testing based on metabolites

The Jaccard Score, precision, recall, and F1 of each compound metabolite was calculated based on SOMs and D-CyPre results were compared with results from CyProduct and BioTransformer.

### Testing results of precision mode

Compared with BioTransformer (Table 5), D-CyPre showed a significant improvement in Jaccard score, precision, and F1 of all subtypes, including a 207% increase in Jaccard score (WAvg), a 286% increase in precision (WAvg), and a 230% increase in F1 (WAvg). However, D-CyPre had a lower recall (WAvg) than BioTransformer, which means that D-CyPre may be a little too cautious when producing metabolites. Compared with CyProduct, D-CyPre achieved a higher Jaccard score and F1 in 2A6, 2C8, 2C9, and 3A4. D-CyPre achieved the highest precision in all subtypes. D-CyPre's precision (WAvg) improved by 31%. However, D-CyPre's Jaccard score (WAvg) and F1 (WAvg) decreased by 2% and 1%, respectively, due to D-CyPre's poor performance in Recall. When considering only four subtypes—2A6, 2C8, 2C9, 34A—compared with CyProduct, the Jaccard score (WAvg) of D-CyPre increased by 24%, precision (WAvg) increased by 56%, and F1 (WAvg) increased by 16%.

In summary, compared to BioTransformer, D-CyPre shows significant improvements in Jaccard score (WAvg), precision (WAvg), and F1 (WAvg) (207%, 286%, 230%). Compared to CyProduct, D-CyPre exhibits an enhancement in precision (WAvg) (31%), indicating more accurate positive results. Additionally, across the four subtypes 2A6, 2C8, 2C9, and 3A4, D-CyPre demonstrates improvements in Jaccard score (WAvg), precision (WAvg), and F1 (WAvg) compared to CyProduct (24%, 56%, 16%).

### Testing results of recall mode

In Recall Mode, although the recall (WAvg) of D-CyPre improved (13%), it was still lower than the other two models (Table 6). Compared with BioTransformer, the Jaccard score, precision, and F1 of D-CyPre were still significantly improved, with their values (WAvg) increasing by 204%, 242%, and 227% respectively. Compared with CyProduct, D-CyPre achieved a higher Jaccard score and F1 for the 1A2, 2A6, 2C8, 2C9, and 3A4 subtypes. Overall, the precision (WAvg) of D-CyPre improved by 16%, while the Jaccard score (WAvg) and F1 (WAvg) decreased by 3% and 2%, respectively. When considering only the

**Table 5 Results (Precision Mode) for nine CYP450 enzymes compared with CyProduct and BioTransformer on 68 reactants of EBoMD2.**

| | 1A2 | 2A6 | 2B6 | 2C8 | 2C9 | 2C19 | 2D6 | 2E1 | 3A4 | WAvg[a] |
|---|---|---|---|---|---|---|---|---|---|---|
| Jaccard score TP/(TP + FP + FN) | | | | | | | | | | |
| D-CyPre | 0.469 | 0.400 | 0.188 | 0.278 | 0.542 | 0.333 | 0.435 | 0.278 | 0.477 | 0.412 |
| CyProduct | 0.471 | 0.391 | 0.333 | 0.263 | 0.429 | 0.385 | 0.511 | 0.611 | 0.366 | 0.420 |
| BioTransformer | 0.113 | 0.263 | 0.200 | 0.152 | 0.128 | 0.125 | 0.121 | 0.125 | 0.112 | 0.134 |
| Precision TP/(TP + FP) | | | | | | | | | | |
| D-CyPre | 0.577 | 0.857 | 0.600 | 0.714 | 0.765 | 0.615 | 0.643 | 0.714 | 0.677 | 0.675 |
| CyProduct | 0.533 | 0.474 | 0.500 | 0.417 | 0.545 | 0.526 | 0.590 | 0.688 | 0.441 | 0.515 |
| BioTransformer | 0.120 | 0.455 | 0.250 | 0.171 | 0.136 | 0.141 | 0.127 | 0.400 | 0.119 | 0.175 |
| Recall TP/(TP + FN) | | | | | | | | | | |
| D-CyPre | 0.714 | 0.429 | 0.214 | 0.313 | 0.650 | 0.421 | 0.574 | 0.313 | 0.618 | 0.526 |
| CyProduct | 0.800 | 0.692 | 0.500 | 0.417 | 0.667 | 0.588 | 0.742 | 0.846 | 0.683 | 0.678 |
| BioTransformer | 0.650 | 0.385 | 0.500 | 0.583 | 0.667 | 0.529 | 0.871 | 0.154 | 0.683 | 0.632 |
| F1 2 × Precision × Recall/(Precision + Recall) | | | | | | | | | | |
| D-CyPre | 0.638 | 0.571 | 0.316 | 0.435 | 0.703 | 0.500 | 0.607 | 0.435 | 0.646 | 0.577 |
| CyProduct | 0.640 | 0.563 | 0.500 | 0.417 | 0.600 | 0.555 | 0.657 | 0.759 | 0.536 | 0.583 |
| BioTransformer | 0.120 | 0.455 | 0.25 | 0.171 | 0.136 | 0.141 | 0.127 | 0.400 | 0.119 | 0.175 |

Note:
[a] The microaverage (weighted average, weighted by the number of metabolites) over the nine CYP450 enzymes.

**Table 6 Results (Recall Mode) for nine CYP450 enzymes compared with CyProduct and BioTransformer on 68 reactants of EBoMD2.**

| | 1A2 | 2A6 | 2B6 | 2C8 | 2C9 | 2C19 | 2D6 | 2E1 | 3A4 | WAvg[a] |
|---|---|---|---|---|---|---|---|---|---|---|
| Jaccard score TP/(TP + FP + FN) | | | | | | | | | | |
| D-CyPre | 0.485 | 0.500 | 0.161 | 0.286 | 0.522 | 0.346 | 0.434 | 0.294 | 0.440 | 0.408 |
| CyProduct | 0.471 | 0.391 | 0.333 | 0.263 | 0.429 | 0.385 | 0.511 | 0.611 | 0.366 | 0.420 |
| BioTransformer | 0.113 | 0.263 | 0.200 | 0.152 | 0.128 | 0.125 | 0.121 | 0.125 | 0.112 | 0.134 |
| Precision TP/(TP + FP) | | | | | | | | | | |
| D-CyPre | 0.571 | 0.692 | 0.227 | 0.545 | 0.800 | 0.600 | 0.500 | 0.833 | 0.635 | 0.598 |
| CyProduct | 0.533 | 0.474 | 0.500 | 0.417 | 0.545 | 0.526 | 0.590 | 0.688 | 0.441 | 0.515 |
| BioTransformer | 0.120 | 0.455 | 0.250 | 0.171 | 0.136 | 0.141 | 0.127 | 0.400 | 0.119 | 0.175 |
| Recall TP/(TP + FN) | | | | | | | | | | |
| D-CyPre | 0.762 | 0.643 | 0.357 | 0.375 | 0.600 | 0.450 | 0.766 | 0.313 | 0.588 | 0.584 |
| CyProduct | 0.800 | 0.692 | 0.500 | 0.417 | 0.667 | 0.588 | 0.742 | 0.846 | 0.683 | 0.678 |
| BioTransformer | 0.650 | 0.385 | 0.500 | 0.583 | 0.667 | 0.529 | 0.871 | 0.154 | 0.683 | 0.632 |
| F1 2 × Precision × Recall/(Precision + Recall) | | | | | | | | | | |
| D-CyPre | 0.653 | 0.667 | 0.278 | 0.444 | 0.686 | 0.514 | 0.605 | 0.455 | 0.612 | 0.573 |
| CyProduct | 0.640 | 0.563 | 0.500 | 0.417 | 0.600 | 0.555 | 0.657 | 0.759 | 0.536 | 0.583 |
| BioTransformer | 0.120 | 0.455 | 0.25 | 0.171 | 0.136 | 0.141 | 0.127 | 0.400 | 0.119 | 0.175 |

Note:
[a] The microaverage (weighted average, weighted by the number of metabolites) over the nine CYP450 enzymes.

above five subtypes, compared with CyProduct, the Jaccard score (WAvg) of D-CyPre increased by 19%, precision (WAvg) increased by 37%, and F1 (WAvg) increased by 12%.

In summary, compared to BioTransformer, D-CyPre demonstrates significant improvements in Jaccard score (WAvg), precision (WAvg), and F1 (WAvg) (204%, 242%, 227%). Moreover, compared to CyProduct, D-CyPre still produces more accurate positive results. Additionally, across five subtypes 1A2, 2A6, 2C8, 2C9, and 3A4, D-CyPre exhibits enhancements in Jaccard score (WAvg), precision (WAvg), and F1 (WAvg) compared to CyProduct (19%, 37%, 12%).

## DISCUSSION

In Precision Mode and Recall Mode, D-CyPre produced good Jaccard scores while maintaining precision and recall values greater than 0.7. In both modes, D-CyPre exhibits significantly higher F1 scores across all subtypes compared to BioTransformer. Additionally, the Jaccard score for all subtypes except 2B6 are significantly higher for D-CyPre compared to BioTransformer in both modes. The 2A6, 2C8, 2C9, and 3A4 subtypes of D-CyPre in Precision Mode and the 1A2, 2A6, 2C8, 2C9, and 3A4 subtypes in Recall Mode all had a higher Jaccard score and F1 than with CyProduct. However, D-CyPre performed worse than CyProduct in other subtypes. Overall, D-CyPre has higher precision than existing models, allowing it to provide more accurate positive results. The results showed that the running time of D-CyPre decreased by 63% and 58% compared with CyProduct and Biotransformer, respectively, when predicting the metabolites of nine CYP450 subtypes of a single compound (Table S14). Additionally, the results indicate that molecular features are necessary to consider in *in silico* metabolism prediction.

The two modes enable this tool to be more flexibly applied in various fields such as drug metabolism research, metabolomics, food, and nutrition studies. When analyzing large amounts of data, using Precision Mode may be more appropriate because it will obtain a small but reliable amount of data from a large amount of data, greatly reducing the workload in subsequent experiments. Conversely, using Recall Mode may be more appropriate for analyzing a small amount of data, as it obtains as much information as possible. For example, a high-throughput study may demand more accurate results, whereas making predictions for several drugs and comparing their corresponding metabolites' mass spectra may require the inclusion of all possibilities.

Feature generators based on two D-MPNN are anticipated to exhibit greater potential compared to those relying on fixed rules as the volume of data increases. This is attributed to the flexibility and specificity of features generated by D-MPNN, which integrate information from molecules, atoms, and bonds. Additionally, when trained on larger-scale datasets, models reliant on fixed rules for feature generation may necessitate manual design of novel features. However, D-CyPre can autonomously extract the requisite features from the datasets. D-CyPre can learn both SOMs based on bonds and SOMs based on atoms at the same time, which makes it fully use existing databases.

Although this study explores the method and effect of a metabolic site prediction model based on D-MPNN and XGBoost, it does not establish a model that can directly produce metabolites, requiring an extra step and more time to further analyze and speculate based

on the results of this tool. In addition, the effects of D-MPNN combined with other models are not discussed in this study, so there may be better combinations. In the future, the effectiveness of combining D-MPNN with other models should be further examined based on the results of this study, and a model that directly generates metabolic products should be established.

## CONCLUSIONS

The present study introduces a novel SOMs identification tool termed D-CyPre. Unlike existing models that generate features for atoms or bonds based on fixed rules, D-CyPre utilizes D-MPNN to generate features for atoms and bonds, thereby enabling better extraction of metabolic-related information. D-CyPre transforms the features of atoms and bonds into a unified dimension and employs a single discriminator for classification, making it adaptable to SOMs defined under various criteria. Additionally, D-CyPre integrates molecular information extracted by D-MPNN with information on atoms and bonds extracted by another D-MPNN to discriminate metabolic sites, thereby further enhancing model accuracy. This model represents the first application of D-MPNN in metabolism prediction and yielded satisfactory _in silico_ outcomes, demonstrating high precision, recall, and Jaccard score. D-CyPre comprises a feature generator and SOMs discriminators and is divided into Precision Mode and Recall Mode. To use the D-CyPre software (as detailed in File S1), a table containing the simplified molecular-input line-entry system (SMILES) strings of all the targeted compounds needs to be input.

### Funding

This work was supported by the National Natural Science Foundation of China (No. 82173957 and 82204599). The State Administration of Traditional Chinese Medicine high-level key discipline of Traditional Chinese medicine (Analysis of Traditional Chinese Medicine) construction project (zyyzdxk-2023265) funded the APC for this article. The funders had no role in study design, data collection and analysis, decision to publish, or preparation of the manuscript.

### Grant Disclosures

The following grant information was disclosed by the authors:
National Natural Science Foundation of China: 82173957 and 82204599.
State Administration of Traditional Chinese Medicine high-level key discipline of Traditional Chinese medicine: zyyzdxk-2023265.

### Competing Interests

The authors declare that they have no competing interests.

### Author Contributions

- Haolan Yang conceived and designed the experiments, analyzed the data, performed the computation work, prepared figures and/or tables, and approved the final draft.

- Jie Liu conceived and designed the experiments, prepared figures and/or tables, authored or reviewed drafts of the article, and approved the final draft.
- Kui Chen performed the experiments, prepared figures and/or tables, and approved the final draft.
- Shiyu Cong performed the experiments, prepared figures and/or tables, and approved the final draft.
- Shengnan Cai performed the experiments, prepared figures and/or tables, and approved the final draft.
- Yueting Li conceived and designed the experiments, authored or reviewed drafts of the article, and approved the final draft.
- Zhixin Jia conceived and designed the experiments, authored or reviewed drafts of the article, and approved the final draft.
- Hao Wu analyzed the data, authored or reviewed drafts of the article, and approved the final draft.
- Tianyu Lou analyzed the data, authored or reviewed drafts of the article, and approved the final draft.
- Zuying Wei analyzed the data, prepared figures and/or tables, and approved the final draft.
- Xiaoqin Yang analyzed the data, prepared figures and/or tables, and approved the final draft.
- Hongbin Xiao conceived and designed the experiments, authored or reviewed drafts of the article, and approved the final draft.

## Data Availability

The code and raw data are available in the Supplemental Files.

## Supplemental Information

Supplemental information for this article can be found online at http://dx.doi.org/10.7717/peerj-cs.2040#supplemental-information.

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
