# Peer review of "D-CyPre: a machine learning-based tool for accurate prediction of human CYP450 enzyme metabolic sites"

_PeerJ Computer Science, doi:10.7717/peerj-cs.2040_

## Round 0.1 · original submission · Major Revisions

The reviewers have substantial concerns about this manuscript. The authors should provide point-to-point responses to address all the concerns and provide a revised manuscript with the revised parts being marked in different color.

**Language Note:** PeerJ staff have identified that the English language needs to be improved. When you prepare your next revision, please either (i) have a colleague who is proficient in English and familiar with the subject matter review your manuscript, or (ii) contact a professional editing service to review your manuscript. PeerJ can provide language editing services - you can contact us at [email protected] for pricing (be sure to provide your manuscript number and title). – PeerJ Staff

Reviewer 1 ·

Basic reporting

The formulas in lines 150, 154 and 156 should be formatted and more rigorously.

Experimental design

The XGBoost method is feasible, but maybe try other methods and then make a comparison of which method will give more accurate predictions.

Validity of the findings

Based on Table 1, more statistical analysis needs to be done to visualize the data.
More examples or article citations are to be given in section 3.1.2.
The authors should consider elaborating on the methodological details, expanding the comparative analysis with existing models, and providing a more thorough discussion on the implications and future directions of their work.

Additional comments

Formatting and layout should be emphasized in the writing of the article and should be consistent throughout the text.

Reviewer 2 ·

Basic reporting

In this manuscript, the authors proposed a method to employs graph neural networks (GNNs) and XGBOOST to predict the metabolic sites of drugs by cytochrome P450 enzymes. It combines information on atoms, bonds, and molecular structure, offering two operational modes - Precision Mode and Recall Mode - to cater to different user requirements. The study demonstrates that D-CyPre outperforms existing models in both validation and test sets. It suggests that incorporating molecular features significantly improves the accuracy of predicting metabolic sites. This result is considered new. I have the following comments for the authors to address before the paper can be accepted for publication in PeerJ Computer Science.

Experimental design

While the paper claims superiority over existing models, a more comprehensive comparison with a wider range of existing tools could strengthen this claim. Detailed comparisons on various metrics, not just accuracy, would be beneficial.

The authors should show the loss curves during training to show that the models are successfully trained.

The t-SNE map is confusing. In line 252-256: “From these results, it can be inferred that there may be two reasons for poor generalization ability of these models on the test set. First, it could be due to an insufficient size of their train sets which leads to some bonds or atoms with similar structures to SOMs in the test set being misclassified as positive while some actual SOMs that are unfamiliar are misclassified as negative.” How can these be interpreted from t-SNE embedding? The authors should elaborate more for the claims.

Validity of the findings

The paper could more thoroughly discuss the limitations of the tool and the specific contexts in which it is most effective. Understanding the scope of the tool's applicability is crucial for end-users.

Additional comments

None

·

Basic reporting

Clear, Professional English: The manuscript is well-written in clear, professional English.
Intro & Background: Provides adequate context and background, highlighting the gap in the existing research.
Literature Reference: Well-referenced, establishing relevance and supporting the research.
Structure & Standards: Conforms to the expected standards and structure, with well-organized sections.
Figures Quality: Figures are relevant, high quality, and well-labeled. I would like to suggest use the resolution as required by the journal because the resolution of figures in PDF seems not high.

Experimental design

Originality & Scope: The research is original and falls within the scope of the journal.
Research Question: The research question is well-defined, relevant, and fills a knowledge gap.
Investigation Rigor: The methods are detailed, allowing for replication, and meet high technical and ethical standards.

Validity of the findings

Data Robustness: The underlying data seem robust, statistically sound, and well-controlled.
Conclusions: The conclusions are directly linked to the research question and are supported by the results.

Additional comments

The manuscript presents a novel tool, D-CyPre, for predicting the metabolic sites of cytochrome P450 enzymes. It offers an in-depth exploration of the methodology and the effectiveness of the tool. The article thoroughly discusses the design, implementation, and validation of the tool, providing a comprehensive understanding of its functionality and advantages over existing models.

I recommend revising certain sentences within the manuscript to enhance clarity and fluency, ensuring they align more closely with fluent English speaker conventions. I have highlighted those sentences in the PDF files with my comments.

---

## Round 0.2 · Minor Revisions

There are some remaining minor revisions that need to be addressed.

Reviewer 2 ·

Basic reporting

The authors addressed my concerns properly and I thus recommend the publication.

Experimental design

NA

Validity of the findings

NA

Additional comments

NA

·

Basic reporting

The flow and writing have been significantly improved in this latest version compared to the previous version.

Experimental design

It provided clear figures of an example of a predictive suite of the D-CyPre model and its design.

Validity of the findings

The authors have developed a new machine learning model aimed at enhancing the prediction of human CYP450 enzyme metabolic sites. Could the authors illustrate how their model outperforms existing models such as CyProduct (Tian et al., 2021), CypReact (Tian et al., 2018), FAME2 (äÌcho et al., 2017), FAME3 (äÌcho et al., 2019a), and BioTransformer (Djoumbou-Feunang et al., 2019) by evaluating all models with the same dataset? This comparison should highlight differences in prediction accuracy within the EXPERIMENTAL RESULTS AND DISCUSSION section. Specifically, please elaborate on the strengths of the new model and identify the shortcomings of other published models in greater detail.

Additionally, it appears that some content in the conclusion section might be more appropriate for the discussion section. I recommend that the conclusion focus primarily on summarizing the study's overall findings and reiterating how the issues or concerns raised in the introduction have been addressed.

I suggest minor revisions in the results, discussion, and conclusion sections to reorganize some content in each section. A separate section for discussion is highly recommended.

---

## Round 0.3 · accepted · Accept

Reviewers are satisfied with the revisions, and I concur to recommend accepting this manuscript.

·

Basic reporting

The manuscript now satisfies the criteria for publication.

Experimental design

The manuscript now satisfies the criteria for publication.

Validity of the findings

The manuscript now satisfies the criteria for publication.